# Ruminal Microbiome Differences in Angus Steers with Differing Feed Efficiencies during the Feedlot Finishing Phase

**DOI:** 10.3390/microorganisms12030536

**Published:** 2024-03-07

**Authors:** Mia N. Carmichael, Madison M. Dycus, Jeferson M. Lourenco, Christina B. Welch, Dylan B. Davis, Taylor R. Krause, Michael J. Rothrock, Francis L. Fluharty, Timothy D. Pringle, Todd R. Callaway

**Affiliations:** 1Department of Animal and Dairy Science, University of Georgia, Athens, GA 30602, USA; mia.carmichael25@uga.edu (M.N.C.); madison.dycus@uga.edu (M.M.D.); jefao@uga.edu (J.M.L.); christina.welch@uga.edu (C.B.W.); dylan.davis@uga.edu (D.B.D.); taylor.krause@uga.edu (T.R.K.); ffluharty@uga.edu (F.L.F.); td.pringle@ufl.edu (T.D.P.); 2Egg Safety and Quality Research Unit, Richard B. Russell Research Center, Agricultural Research Service, United States Department of Agriculture, Athens, GA 30605, USA; michael.rothrock@usda.gov

**Keywords:** beef cattle, feed efficiency, methane, *Methanobrevibacter*, microbiota

## Abstract

The catabolic activity of the ruminal microbial community of cattle enables the conversion of low-quality feedstuffs into meat and milk. The rate at which this conversion occurs is termed feed efficiency, which is of crucial importance given that feed expenses account for up to 70% of the cost of animal production. The present study assessed the relationship between cattle feed efficiency and the composition of their ruminal microbial communities during the feedlot finishing period. Angus steers (*n* = 65) were fed a feedlot finishing diet for 82 days and their growth performance metrics were evaluated. These included the dry matter intake (DMI), average daily gain (ADG), and residual feed intake (RFI). Steers were rank-ordered based upon their RFI, and the five lowest RFI (most efficient) and five highest RFI (least efficient) steers were selected for evaluations. Ruminal fluid samples were collected on days 0 and 82 of the finishing period. Volatile fatty acids (VFA) were quantified, and microbial DNA was extracted and the 16S rRNA gene was sequenced. The results showed that the ADG was not different (*p* = 0.82) between efficiency groups during the 82-day feedlot period; however, the efficient steers had lower (*p* = 0.03) DMI and RFI (*p* = 0.003). Less-efficient (high RFI) steers developed higher (*p* = 0.01) ruminal *Methanobrevibacter* relative abundances (*p* = 0.01) and tended (*p* = 0.09) to have more *Methanosphaera*. In high-efficiency steers (low RFI), the relative abundances of *Ruminococcaceae* increased (*p* = 0.04) over the 82-day period. The molar proportions of VFA were not different between the two efficiency groups, but some changes in the concentration of specific VFA were observed over time. The results indicated that the ruminal microbial populations of the less-efficient steers contained a greater relative abundance of methanogens compared to the high-efficiency steers during the feedlot phase, likely resulting in more energetic waste in the form or methane and less dietary energy being harvested by the less-efficient animals.

## 1. Introduction

The dietary composition of ruminant animals such as cattle, along with the microbial community structure of the rumen, influence animal productivity. The activity of the resident ruminal microbial consortium allows ruminant animals to occupy a specific ecological niche as consumers of cellulosic fiber [1]. Because mammals do not produce fiber-degrading enzymes, ruminants are entirely dependent on the degradative capacity of the ruminal bacteria, fungi, and protozoa to ferment cellulose and hemicellulose into usable energy for the host [2,3,4]. However, dependence on microbial fermentation means that ruminant animals exchange efficiency for substrate flexibility. The ruminal microbial fermentation of structural carbohydrates produces metabolizable energy for the host animal in the form of volatile fatty acids (VFA); however, fermentation byproducts, such as methane, can represent a loss of up to 12% of the dietary energy [5,6,7].

Feed costs are the largest variable expense associated with producing beef cattle [8], accounting for approximately 70% of the cost of producing finished cattle [9]. Beef production in the U.S. utilizes feedlots at the end of the production chain, where a predominantly grain diet is fed to ensure efficient muscle and bone growth [10]. When cattle are fed a grain-based diet, energy density is increased and therefore cattle production is optimized in a well-managed intensive feedlot system [11]. However, when not well managed, pushing production efficiency can have negative impacts on animal health. The excessive consumption of fermentable carbohydrates by ruminant animals may lead to acidosis as acids accumulate in the rumen, causing a reduction in pH and potential damage to the gastrointestinal wall [12]. To prevent acidosis via management practices, a moderate meal size and the addition of dietary roughage allow grains to be processed adequately [12]. Cereal grains are a major energy source for cattle due to their high concentrations of starch [11], a component that is well digested in the gastrointestinal tract when adequately processed [13]. Previous studies have shown a close association between the ruminal microbiota composition, fermentation end products, and cattle feed efficiency [14,15,16]. In addition to feed management and the gastrointestinal microbiome, feed efficiency can be influenced by breed, body composition, age, and the environment [17]. Improving the cattle feed efficiency could decrease both feeding costs and environmental impacts, such as decreased the production of methane in beef production systems [14].

The Residual Average Daily Gain (RADG) is a feed efficiency selection tool adopted by the American Angus Association in their breeding selection programs because it has a high degree (~0.3) of heritability [18,19]. Selection based upon RADG allows producers to improve the feedlot efficiency in finishing cattle by focusing on predictors of a sire’s genetic capacity for their future progeny to have good postweaning gain relative to other sires, assuming that a constant amount of feed is consumed. Therefore, the present study was designed to compare the composition and end products of the ruminal microbiome of beef cattle with different feed efficiencies at the beginning and end of an 82-day feedlot phase. We hypothesized that animals with different feed efficiencies would undergo differential changes in their microbial community compositions over the course of the finishing period.

## 2. Materials and Methods

### 2.1. Animals, Experimental Design, and Management

All procedures involving animals were verified and approved by the University of Georgia’s Office of Animal Care and Use (AUP #A2012 11-006-R1). The present study was conducted to extend the scope of a previous study [20]. Briefly, commercial Angus cows were bred to high- and low-efficiency bulls to determine the effect of selection (i.e., RADG and carcass marbling on expected progeny differences [EPDs]) on the productivity, performance, and carcass quality of this selected herd. The current study utilized the fifth generation of steers born to that selection program and analyzed their performance in a commercial grain-fed feedlot finishing system. The steers (*n* = 65; approximately 13 months of age) were fed in a commercial feedlot (Ridgefield Farm L.L.C.; Brasstown, NC, USA; 35.0391° N, 83.9576° W) from 8 March 2018 to 29 May 2018. Steers were rank-ordered based upon their residual feed intake (RFI) data collected in the feedlot period, where a lower RFI is more desirable. From the RFI rankings, the 5 most efficient (*n* = 5; body weight 503 ± 65 kg; approximately 13 months of age) and 5 least efficient (*n* = 5; body weight 512 ± 47 kg; approximately 13 months of age) steers were selected for the present analysis of their ruminal microbiomes and volatile fatty acids (VFAs).

Steers were adapted to the high-grain feedlot finishing ration for 21 d prior to the trial, and all rations were formulated to meet the nutrient requirements for finishing cattle shown in Table 1 [21]. Over the 82 d feeding trial, the individual daily feed intake was recorded using a GrowSafe feed intake monitoring system (GrowSafe Systems^®^ Ltd., Calgary, AB, Canada) [22,23,24]. Steers were weighed at the beginning, mid-point, and end of the feedlot experimental period (d 0, 41, and 82, respectively). Using the individual dry matter intakes (DMIs) and body weight gains for each steer, the RFI was calculated as the difference between the actual and expected DMI. The expected DMI was the estimate for each steer, following the linear regression of the midpoint, metabolic body weight and average daily gain (ADG) on the actual DMI. 

### 2.2. Sample Collection, DNA Extraction and Sequencing

Prior to morning feedings, ruminal samples were collected upon entering (d 0) and exiting (d 82) the feedlot finishing period via esophageal tubing, as described in detail elsewhere [25]. All samples were immediately placed on wet ice and transported to the laboratory, where they were stored at −20 °C until further processing. The same ruminal samples per animal were used for all analysis. DNA was extracted from the rumen fluid samples according to a semi-automated extraction protocol, which utilizes a combinatorial approach of enzymatic and mechanical DNA extraction methods [26]. Briefly, the procedure started with 0.5 mL of homogenized rumen fluid. The mechanical portion of the protocol used a FastPrep 24 Instrument (MP Biomedicals LLC, Irvine, CA, USA) to disrupt the cells. The enzymatic portion used InhibitEX^®^ Tablets (QIAGEN, Venlo, The Netherlands), and proteinase K. elution and the purification of DNA were performed using an automated robotic workstation (QIAcube; QIAGEN, Venlo, The Netherlands). The determination of the DNA concentration and purity was performed spectrophotometrically using the Synergy™ H4 Hybrid Multi-Mode Microplate Reader (BioTek Instruments Inc.; Winooski, VT, USA). After DNA extraction, samples were taken to the Georgia Genomics and Bioinformatics Core facility for library preparation and 16S rRNA gene sequencing. The library preparation included PCR replications using the forward: S-D-Bact-0341-b-S-17 (5′-CCTACGGGNGGCWGCAG-3′) and reverse: S-D-Bact-0785-a-A-21 (5′-GACTACHVGGGTATCTAATCC-3′) primer pair [27]. Samples were sequenced using the Illumina MiSeq system (Illumina Inc., San Diego, CA, USA). All files are publicly available from the MG-RAST website, under accession number 4870427.3.

### 2.3. Volatile Fatty Acid Determination

Rumen samples were analyzed for VFA using the methodology described in Lourenco et al. [28]. Briefly, 2 mL of each sample was centrifuged at 10,000× *g* for 10 min and 1 mL of the supernatant was mixed with 0.2 mL of 25% (*wt*/*vol*) meta-phosphoric acid, vortexed and frozen overnight. Subsequently, the samples were thawed and centrifuged (10,000× *g*, 10 min), before 1 mL of supernatant was transferred to screw-thread vials that contained 2 mL of ethyl acetate. The samples were vortexed and allowed to settle for at least 5 min. The upper layer was transferred (1 mL) into gas chromatography vials for VFA analysis using a Shimadzu GC-2010 Plus gas chromatograph (Shimadzu Corporation, Kyoto, Japan) equipped with a flame ionization detector and capillary column (Zebron ZB-FFAP; 30 m × 0.32 mm × 0.25 µm; Phenomenex Inc., Torrance, CA, USA). The sample injection volume was 1.0 µL, and helium was used as the carrier gas. The starting temperature of the column was set at 110 °C and gradually increased to 200 °C, the injector temperature was set at 250 °C, and the detector temperature was set at 350 °C. The output variables recorded were the acetate, propionate, isobutyrate, butyrate, isovalerate, valerate, and caproic acid levels.

### 2.4. Sequencing Data

Sequencing data were demultiplexed and downloaded as FASTQ files. Pair-end reads were merged using BBMerge Paired Read Merger v37.64, and the files were further analyzed using the QIIME pipeline v1.9.1 [29]. The files were quality filtered and then converted into FASTA files. The sequences were clustered into operational taxonomic units (OTUs) at 97% similarity and compared to the Greengenes database. A sampling depth of 28,662 sequences per sample was used for the diversity analyses. The computed alpha-diversity indexes were the number of observed OTUs, Chao1, species evenness, Shannon index, and Faith’s phylogenetic diversity.

### 2.5. Statistical Analyses

Statistical analyses were performed using the software Minitab^®^ v18.1. The alpha-diversity indexes and bacterial relative abundances were analyzed by ANOVA within each individual group of steers (i.e., low vs. high RFI), and the feedlot period (beginning or end) was used as a factor. The animal performance data were analyzed across the RFI groups (high or low). For all data analyzed, Fisher’s Least Significant Difference (LSD) test was performed after ANOVA. Results were considered significant when *p* ≤ 0.05 and were categorized as trends when 0.05 < *p* ≤ 0.10.

## 3. Results

### 3.1. Steer Growth and Efficiency Performance

The performance and efficiency of each steer were individually assessed throughout the 82 d feedlot period by measuring the dry matter intake (DMI), average daily gain (ADG), feed/gain ratio, and residual feed intake (RFI; Table 2). The most efficient steers (low RFI) had a lower average DMI (*p* = 0.03) than the least efficient (high RFI) steers. In addition, the average difference in RFI between the least and most efficient groups was 1.85 kg DMI/day (*p* = 0.003). 

### 3.2. Volatile Fatty Acid Concentrations

The concentrations of most VFAs, as well as the total VFA concentrations, increased between d 0 and d 82 of the feedlot period (Table 3). Changes in the total VFA concentrations in the ruminal samples over the feedlot period were greater in the less efficient steers than in the efficient steers. The propionate, butyrate, valerate, and total VFA concentrations were increased from d 0 to d 82 in inefficient steers (*p* ≤ 0.03). Meanwhile, only the isobutyrate and isovalerate concentrations were increased (*p* = 0.01) in inefficient steers from d 0 to d 82. 

### 3.3. Microbial Community Structure

The number of observed OTUs increased numerically in both the low and high RFI groups during the 82 d feedlot period; however, those differences were not statistically significant (*p* ≥ 0.14; Table 4). Chao1 increased in both groups of steers during the feedlot phase, especially in the high-RFI steers, where a tendency for greater microbial richness was observed (*p* = 0.07). The microbial diversity, when assessed by the Shannon index, revealed a trend of increased diversity in the rumen of high-RFI steers (*p* = 0.10) over the course of the feedlot phase; however, when the microbial diversity was expressed as Faith’s Phylogenetic Diversity index, no differences were observed (*p* = 0.21) in this group of animals. In the most efficient steers (low RFI), no differences (*p* ≥ 0.20) in the microbial diversity were observed during the feedlot finishing phase, regardless of the metrics used. 

An analysis of the rumen fluid samples identified 10 bacterial phyla with a relative abundance equal or greater than 0.2% in both high-RFI (low efficiency) and low-RFI (high efficiency) steers (Table 5 and Table 6). The relative abundance of the predominant phyla varied over the course of the 82 d feedlot period, as well as between the high- and low-RFI steers. At the beginning of the feedlot period, *Bacteroidetes*, *Firmicutes*, and *Actinobacteria* were the most predominant phyla for both high- (Table 5) and low-RFI (Table 6) steers, accounting for more than 90% of the total OTU in these steers. However, at the end of the feedlot period (d 82), the top three phyla were *Firmicutes*, *Bacteroidetes* and *Proteobacteria*, which collectively composed more than 88% of the OTU.

The relative abundance of *Firmicutes* was higher at the end of the finishing period compared to the beginning in the most efficient animals (i.e., low RFI; *p* = 0.02; Table 6) and tended to increase (*p* = 0.08) in the least efficient animals (high RFI; Table 5). Additionally, there was a trend (*p* = 0.09) for ruminal *Bacteroidetes* to be lower in abundance in the least efficient animals during the feedlot period. The increase in *Firmicutes:Bacteroidetes* ratio was significant (*p* = 0.05) in the least efficient steers (Table 5) and tended (*p* = 0.10) to increase in the more efficient steers (Table 6) from the beginning to the end of the feeding trial.

*Prevotellaceae* was the most prevalent family identified in both groups of steers at the beginning of the feedlot period, with relative abundances averaging approximately 38% of the total bacterial community for RFI groups (Figure 1). However, at the end of the feedlot period, *Prevotellaceae* was significantly lower (*p* = 0.05) in the rumen of both high- and low-RFI steers (at 14 and 20%, respectively; Figure 1). *Ruminococcaceae* had the greatest increase in the most efficient animals (Figure 1; *p* = 0.04); meanwhile, in the least efficient animals, this increase was smaller (*p* = 0.17). More changes occurred at the family level throughout the feedlot period in the least efficient (high-RFI) steers: the abundance of *Erysipelotrichaceae* decreased (d 0: 1.2%, d 82: 0.7%; *p* = 0.01), whereas the family *Methanobacteriaceae* increased (*p* = 0.01) in abundance (d 0: 1.2%, d 82: 2.0%). 

Two archaeal genera were identified in the rumen of the steers: *Methanobrevibacter* and *Methanosphaera* (Figure 2). The archaeal phylum *Euryarchaeota* was significantly higher (*p* = 0.01) at the end of the finishing period than at the beginning in the least efficient steers. The increase in the abundance of *Euryarchaeota* subsequently resulted in an increase in the methanogenic genera *Methanobrevibacter* and *Methanosphaera* within this phylum (Figure 2). In the least efficient steers (high-RFI), the *Methanobrevibacter* relative abundance increased (*p* = 0.01; Figure 2A) and the *Methanosphaera* relative abundance tended to increase (*p* = 0.09; Figure 2C) during the 82 d feedlot period. However, in the most efficient (low RFI) steers, the *Methanobrevibacter* and *Methanosphaera* abundances were not different (*p* ≥ 0.83) over the 82 d feedlot period. Thus, during the 82 d feedlot period, there was a developmental increase in these two methanogenic archaea within the rumen of steers characterized as least efficient (high RFI). Conversely, no methanogenic archaeal changes were detected in the rumen of highly efficient steers (low RFI). 

## 4. Discussion

Feedlot cattle are fed high grain rations to maximize their rate of gain, to increase intramuscular marbling and improve carcass quality [11,12,13,30]. Grain diets are commonly high in starch, a readily fermentable substrate that can alter the microbial community composition of the rumen [11,12,13,31]. The present study’s findings showing that the most efficient steers (low RFI) had a lower average DMI and RFI than the least efficient (high RFI) steers were previously demonstrated in both dairy and beef cattle [32]. 

Since the changes in the total VFA concentrations in the ruminal samples over the feedlot period were more pronounced in the less efficient steers, this may reflect their inability to quickly absorb these compounds. However, since the present study did not quantify VFA absorption, this hypothesis cannot be supported. Overall, the VFA concentrations were dominated by acetate regardless of the feed efficiency status. We had hypothesized that we would observe differences in the ruminal VFA concentrations between the efficiency groups that could be attributed to differences in the microbiome composition; however, the differences observed in the ruminal VFA concentrations were potentially masked by differences in absorption across the ruminal epithelium.

The number of observed OTUs increased numerically in both groups of steers during the 82 d feedlot period. Similarly, the Chao1 index, which estimates microbial richness while accounting for rare species [33], increased in both groups of steers during the feedlot phase, especially in high-RFI steers. The microbial diversity, when assessed through the Shannon index, revealed a trend of increased diversity in the rumen of high-RFI steers over the course of the feedlot phase; however, when the microbial diversity was expressed using Faith’s Phylogenetic Diversity index, no differences were observed, indicating that the alpha-diversity metric used can affect the conclusions drawn. In the most efficient steers, no differences in the microbial diversity were observed during the feedlot finishing phase, regardless of the metrics used. 

The relative abundance of *Firmicutes* was higher at the end of the finishing period compared to the beginning in the most efficient animals (i.e., low RFI). A similar trend was observed in the inefficient animals (high-RFI), although the increase was not as significant. Conversely, ruminal *Bacteroidetes* tended to decrease in abundance in the least efficient animals during the feedlot period. However, the most noteworthy difference was seen in an archaeal phylum with low ruminal abundance, namely *Euryarchaeota*. It significantly increased at the end of the finishing period in the least efficient steers. The phylum *Euryarchaeota* contains a diverse group of obligate anaerobic methanogens [34]. Therefore, an increase in the abundance of *Euryarchaeota* subsequently resulted in an increase in the methanogenic genera *Methanobrevibacter* and *Methanosphaera* within this phylum.

A classic study by Turnbaugh and collaborators identified a link between the *Firmicutes*/*Bacteroidetes* ratio and obesity in mice [35], indicating that this ratio affected the efficiency with which those animals harvested energy from their diets. It should be noted that the importance of this ratio in terms of microbiome activity and host physiology is still poorly understood [36]. However, the increase in the *Firmicutes*/*Bacteroidetes* ratio during the 82 d feedlot period suggests that in both efficiency groups the ruminal microbial community structure changed to harvest more energy from the readily fermentable starch-rich diet.

The family *Prevotellaceae* consists of amylolytic Gram-negative bacteria that proliferate rapidly in the presence of high-starch diets, such as feedlot diets [37,38,39]. Thus, the decrease in *Prevotellaceae* observed in both efficiency groups during the feedlot phase was unexpected. However, the *Ruminococcaceae* family contains bacteria involved in the degradation of complex carbohydrates, including neutral detergent fiber (NDF) degradation [40,41], so the increased abundance of *Ruminococcaceae* in both groups of steers could be partially explained by environmental niche-filling caused by the decrease in *Prevotellaceae*. Moreover, the increase in *Ruminococaceae* could potentially mitigate the effects of the low ruminal pH associated with the starch-containing feedlot diet by fermenting dietary carbohydrate to VFA rather than lactate [42,43].

While more changes occurred at the family level throughout the feedlot period, these changes occurred in bacterial families with smaller relative abundances and only occurred in the least efficient steers. For example, the abundance of *Erysipelotrichaceae* decreased, whereas the family *Methanobacteriaceae* increased in abundance in the least efficient steers. Previous studies have shown a greater abundance of the *Erysipelotrichaceae* family in low-methane-producing animals, with less hydrogen being produced; this leads to less interspecies hydrogen transfer and less methane production [44]. It was hypothesized that the enrichment of the *Erysipelotrichaceae* family was associated with the higher ruminal turnover rates found in low-methane-emitting ruminants, and that the more rapid ruminal liquid dilution rate favored rapid homofermentative starch-degrading microorganisms (e.g., *Streptococcus bovis*, *Sharpea azabuensis*) [45,46], which might have been the case in our study.

In addition to VFA, the ruminal microbial fermentation process produces hydrogen, carbon dioxide, and methane, which are not utilized by the host [1,47,48]. Methane is a potent greenhouse gas produced by ruminal archaea and can represent a loss of up to 12% of the dietary energy provided to ruminant animals [5,49,50]. Archaea are metabolically diverse organisms that in the rumen are often associated with bacteria, fungi, and protozoa [51], and that often serve as a reducing equivalent sink for NADH disposal under highly reduced ruminal environment conditions [52,53]. Interspecies hydrogen transfer was demonstrated using co-cultures of *Ruminococcus albus* and *Methanobacterium ruminantium*, which caused the fermentation end products to shift from ethanol to acetate, enabling *R*. *albus* to receive a higher ATP yield [54,55,56]. In our study, the increase in ruminal *Methanobrevibacter* and *Methanosphaera* abundances in the least efficient steers (high-RFI) suggests that there was potentially a greater waste of dietary energy in the form of methane in these steers, contributing to their reduced feed efficiency.

Shifts in the relative abundances of *Methanobrevibacter* and *Methanosphaera* have been correlated with a decrease in the abundance of *Thermoplasmata* [57], which are methanogens that can utilize methanol and methylamine as a substrate [58]. In the current study, the abundance of *Thermoplasmata* was negligible (less than 0.00002% on average), and many animals did not even have OTUs assigned to this taxa, limiting our ability to run similar correlations. Although this relatively small microbial community of archaea might not be the only factor that underlies shifts in cattle efficiency and methane production, further examinations should be conducted to assess the impacts of *Thermoplasmata* on rumen fermentation.

Changes in ruminal microbial communities have been associated with feed efficiency differences in beef cattle [15,59,60]. More recently, the linkages between rumen microbiome activities and feed efficiencies in beef cattle have been studied by utilizing advanced metagenomics and metabolomics approaches [61,62]. While the RFI was used to define feed efficiency in the present study, a potential problem arises when the RFI is used due to differences in feed intake, which can greatly influence the composition of the microbiota [63]. To avoid changes associated with feed intake, metabolomics has been used to assess the effect of ruminal metabolites on ADG. Artgegoitia et al. [63] indicated that 33 individual metabolites were associated with differences in ADG in beef cattle. The results suggested that the balance between microbial community activity and the ruminal absorption of VFA (and other acids) impacted the ADG of crossbred beef steers. More recently, de Almeida et al. [64] utilized untargeted metabolomics to provide a snapshot of the rumen fluid metabolome and observed 1882 molecular features with 67 molecular features, including the following: amino acids, dicarboxylic acids, carboxylic acids, lactones, fatty acids derivatives and indole compounds. The production traits of beef cattle have been improved through animal selection using traditional quantitative genetics approaches [65]; however, there is still a need to understand the physiological mechanisms that contribute to variations in feed efficiency, including the impact of changes in the composition of the ruminal microbial community. While the complexity of the rumen remains vast and not well understood, these findings elucidate the potential of the rumen as a reservoir of novel metabolites and enzymatic activities that can be used to improve sustainability and reduce the global footprint of animal agriculture.

## 5. Conclusions

Our results suggest that steer feed conversion is impacted by shifts in the composition of the ruminal microbial community. Despite the limitations posed by the sample size in the present study, we observed an increase in the abundance of the family *Ruminococcaceae* in the rumen of the most efficient steers over the course of the finishing phase. This could result in a more stable pH, potentially resulting in more energy harvested from the diet by microbial fermentation and thereby increasing steer efficiency. Conversely, the less efficient steers developed a higher relative abundance of the methanogenic archaea *Methanobrevibacter* and *Methanosphaera* over the course of the feedlot period, suggesting that a greater loss of energy as methane might have occurred in the rumen of these steers. Further studies are required to investigate the less abundant microbial community members and to understand the hindgut microbiome and associated holistic interactions more deeply. Additionally, the further integration of metabolomic approaches is necessary to understand how shifts in the composition of the ruminal microbial community affect feed efficiency in beef cattle. While the current study provides valuable insights, it is important to note the limitation posed by its sample size, so further studies should employ larger sample sizes to enhance the robustness of the findings. However, as we develop a more comprehensive understanding of the gastrointestinal microbiome and metabolome interactions, further improvements can be made to reduce production costs, improve management strategies, and increase energetic efficiency.

## Figures and Tables

**Figure 1 microorganisms-12-00536-f001:**
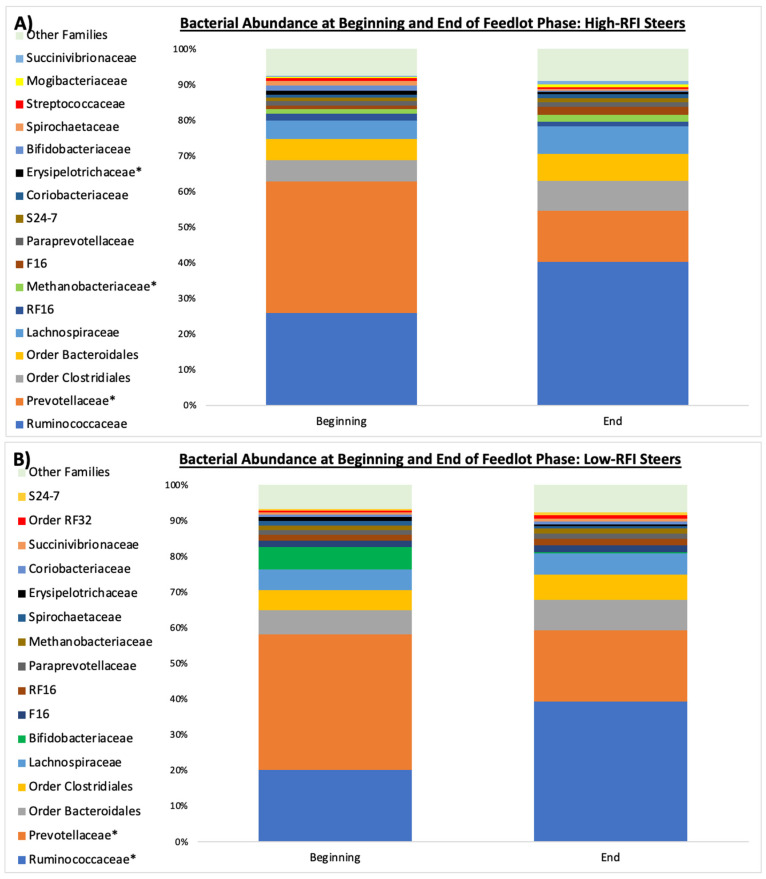
Relative abundance of microorganisms at the family level in ruminal samples collected at the beginning (d 0) and end (d 82) of the feedlot period in high-RFI Angus steers (*n* = 5) fed in a commercial feedlot (**A**) and low-RFI Angus steers (*n* = 5) (**B**). * Indicates significant differences (*p* ≤ 0.05).

**Figure 2 microorganisms-12-00536-f002:**
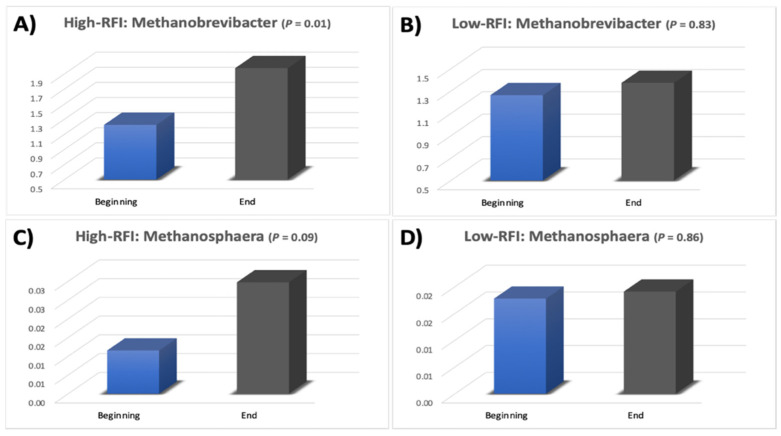
Relative abundance of *Methanobrevibacter* and *Methanosphaera* in ruminal samples (*n* = 5 for high RFI and 5 for low RFI) collected at the beginning (d 0) and end (d 82) of the feedlot period in high-RFI steers (**A**,**C**) and low-RFI steers (**B**,**D**) fed in a commercial feedlot.

**Table 1 microorganisms-12-00536-t001:** Composition of the diets fed to Angus steers (*n* = 65) during the transition and finishing periods of the feedlot trial. Transition diet was fed from d 0–21 prior to the finishing diet, which was fed for d 22–82.

	Transition Diet	Finishing Diet
Ingredient, % DM		
Corn	41.12	56.20
Dried distillers grains	22.18	19.54
Corn gluten feed	-	7.08
Soybean hulls	15.80	-
Barley straw	6.15	4.36
Vitamin/Mineral Premix	4.47	4.76
Corn silage	10.27	8.05
Total	100.00	100.00
Nutrient, % DM		
Dry Matter, %	62.00	62.00
Feedlot NEm, Mcal/cwt	91.84	95.17
Feedlot NEg, Mcal/cwt	62.10	65.00
Crude Protein, %	14.63	14.51
Roughage, %	16.43	12.40
Rough NDF, %	9.28	6.92
Fat, %	5.11	5.28
Calcium, %	0.75	0.70
Phosphorus, %	0.39	0.45
Potassium, %	0.90	0.71
Magnesium, %	0.22	0.21
Sulfur, %	0.25	0.26
Trace Mineral Salt, %	0.21	0.22

**Table 2 microorganisms-12-00536-t002:** Animal performance observed during the finishing period (d 0–82) according to the Residual Feed Intake (RFI) classification in feedlot Angus steers (*n* = 5/group).

	RFI Classification			
Item	High	Low	SEM	*p*-Value ^1^
Average body weight, kg ^2^	545.7	564.0	14.61	0.62
Dry matter intake (DMI), kg/day	13.02	10.89	0.52	0.03
Residual feed Intake (RFI), kg	0.76	−1.09	0.37	0.003
Feed/gain ratio, kg	12.43	11.27	0.61	0.37
Average daily gain (ADG), kg/day	1.05	1.02	0.07	0.82

^1^ *p*-value for the contrast between high and low-RFI steers. ^2^ Average live weight during the 82-day finishing period.

**Table 3 microorganisms-12-00536-t003:** Volatile fatty acid (VFA) concentrations detected in the rumen at the beginning (d 0) and end (d 82) of the feedlot period in steers classified by RFI as either efficient (*n* = 5) or efficient (*n* = 5).

	VFA Concentration (mM)			
Item	Beginning	End	SEM	*p*-Value ^1^
	Less-efficient steers (high RFI)			
Acetate	50.9	69.5	5.10	0.06
Propionate	16.5	23.7	1.76	0.03
Butyrate	11.1	16.8	1.27	0.01
Isobutyrate	0.8	1.1	0.09	0.06
Valerate	1.0	1.5	0.12	0.03
Isovalerate	2.1	2.9	0.28	0.19
Caproate	0.3	0.4	0.09	0.79
Total VFA	82.8	115.8	8.24	0.03
	More-efficient steers (low RFI)			
Acetate	56.2	64.8	3.38	0.22
Propionate	21.0	22.4	1.66	0.70
Butyrate	14.1	15.9	1.13	0.47
Isobutyrate	0.7	1.1	0.07	0.01
Valerate	1.1	1.5	0.13	0.11
Isovalerate	1.9	3.0	0.22	0.01
Caproate	0.2	0.2	0.04	0.38
Total VFA	95.1	108.8	5.78	0.26

^1^ *p*-value for the contrast between beginning (d 0) and end of the feedlot finishing period (d 82).

**Table 4 microorganisms-12-00536-t004:** Alpha diversity metrics observed during the 82 d finishing period according to the Residual Feed Intake (RFI) classification from Angus steers (*n* = 5/group).

		Feedlot Finishing Period		
Item	Beginning (d 0)	End (d 82)	SEM	*p*-Value ^1^
High-RFI Steers (*n* = 5)				
Number of OTUs	1,587	1,732	54.20	0.14
Chao1	2,366	2,575	75.27	0.07
Faith’s Phylogenetic Diversity	92.1	97.6	2.65	0.21
Shannon Index	7.19	7.89	0.18	0.10
Low-RFI Steers (*n* = 5)				
Number of OTUs	1,498	1,787	114.0	0.21
Chao1	2,284	2,727	177.59	0.22
Faith’s Phylogenetic Diversity	89.0	100.8	4.73	0.20
Shannon Index	7.14	7.74	0.32	0.41

^1^ *p*-value for the contrast between beginning (d 0) and end of the feedlot finishing period (d 82).

**Table 5 microorganisms-12-00536-t005:** Relative abundance of microorganisms * (phylum level) observed at the beginning (d 0) and end (d 82) of the finishing period for high-residual-feed-intake (RFI) Angus steers (*n* = 5 steers) in a commercial feedlot.

	Feedlot Finishing Period		Average Abundance	
Bacterial Phyla	Beginning	End		*p*-Value ^1^
*Firmicutes*	41.34	60.37	50.86	0.08
*Bacteroidetes*	47.34	25.98	36.66	0.09
*Actinobacteria*	3.01	1.56	2.28	0.19
*Proteobacteria*	0.97	2.27	1.62	0.12
*Euryarchaeota*	1.24	2.01	1.62	0.01
TM7	1.05	2.12	1.59	0.09
*Spirochaetes*	1.40	0.53	0.97	0.06
*Tenericutes*	0.32	0.89	0.61	0.14
*Cyanobacteria*	0.49	0.44	0.46	0.81
*Planctomycetes*	0.35	0.39	0.37	0.82
Other Phyla	2.49	3.43	2.96	0.07
*Firmicutes*/*Bacteroidetes* ratio	1.14	2.80	1.97	0.05

* Only phyla with average abundance ≥ 0.2% are shown. ^1^ *p*-value for the contrast between beginning and end of the feedlot finishing period.

**Table 6 microorganisms-12-00536-t006:** Relative abundance of microorganisms * (phylum level) observed at the beginning (d 0) and end (d 82) of the finishing period for low-residual-feed-intake (RFI) Angus steers (*n* = 5 steers) in a commercial feedlot.

	Feedlot Finishing Period		Average Abundance	
Bacterial Phylum	Beginning	End		*p*-Value ^1^
*Firmicutes*	34.95	55.14	45.04	0.02
*Bacteroidetes*	48.73	33.04	40.88	0.12
*Actinobacteria*	7.21	1.29	4.25	0.34
TM7	1.81	2.06	1.94	0.85
*Proteobacteria*	1.48	2.31	1.90	0.43
*Euryarchaeota*	1.28	1.39	1.34	0.83
*Spirochaetes*	1.31	0.81	1.06	0.42
*Cyanobacteria*	0.39	0.33	0.36	0.74
*Tenericutes*	0.16	0.46	0.31	0.20
*Planctomycetes*	0.21	0.27	0.24	0.55
Other Phyla	2.46	2.91	2.69	0.55
*Firmicutes*/*Bacteroidetes* ratio	0.78	1.96	1.37	0.10

* Only phyla with average abundance ≥ 0.2% are shown. ^1^ *p*-value for the contrast between beginning and end of the feedlot finishing period.

## Data Availability

The raw data supporting the conclusions of this article will be made available by the authors on request.

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
