# Peer review of "Ruminal Microbiome Differences in Angus Steers with Differing Feed Efficiencies during the Feedlot Finishing Phase"

_microorganisms, 2024, doi:10.3390/microorganisms12030536_

Round 1
Reviewer 1 Report
Comments and Suggestions for Authors
This research is very interesting, and introducing the Ruminal Microbiome Differences in Angus Steers with Differing Feed Efficiencies During the Feedlot-Finishing Phase, but there is still some problems,
1 The background introduction is insufficient, and more recent research progress should be supplemented to highlight the importance and novelty of this study.
2 The image in the paper is not clear and the pixels are too low. Please ask the author to improve the image again.
3 The conclusion section is not concise enough, please ask the author to provide a clear conclusion.
In summary, this paper is a highly novel and valuable paper that is worthy of publication in this journal.
Author Response
Reviewer 1
-
This research is very interesting, and introducing the Ruminal Microbiome Differences in Angus Steers with Differing Feed Efficiencies During the Feedlot-Finishing Phase, but there is still some problems,
1 The background introduction is insufficient, and more recent research progress should be supplemented to highlight the importance and novelty of this study.
- Response. The reviewer is correct, we have adjusted the background information and included several new references that highlight the novelty and importance of this study. We thank the reviewer for their constructive point
-
2 The image in the paper is not clear and the pixels are too low. Please ask the author to improve the image again.
- We have changed them into JPG figures
-
3 The conclusion section is not concise enough, please ask the author to provide a clear conclusion.
-
The reviewer is correct. We added "
While the current study provides valuable insights, it is important to note its limitation of sample size, so further studies should employ larger sample sizes to enhance the robustness of the findings. However, as we develop a more comprehensive understanding of the gastrointestinal microbiome and metabolome interactions, further improvements can be made to reduce production costs, improve management strategies, and increase energetic efficiency."
-
In summary, this paper is a highly novel and valuable paper that is worthy of publication in this journal.
- We thank the reviewer for their kind words and the effort they put in to improving this manuscript.
Reviewer 2 Report
Comments and Suggestions for Authors
The study lacks some results to be complete.
The purpose of the study is not clear. Please clarify it.
Were rumen fluid samples drawn from all animals? Explain this.
Separate the Sequencing data part from the statistical analysis part.
The hypothesis mentioned in line 65 regarding the study is intuitive, so what is new that the study offers?
What were the weights of the animals that started the experiment, what were their weights at the end of the experiment, and were they affected by the final result?
Author Response
Reviewer 1
- Comment: The study lacks some results to be complete.
- Answer: We recognize the limitations of our study; however, we reported the results we had. If the reviewer can be more specific, we might be able to improve the manuscript accordingly.
- Comment: The purpose of the study is not clear. Please clarify it.
- Answer: We stated the purpose of the study at the end of our introduction. It reads: . “Therefore, the present study was designed to compare the composition and end products of the ruminal microbiome of beef cattle with different feed efficiencies at the beginning and end of an 82-day feedlot phase. We hypothesized that animals with different feed efficiencies would undergo differential changes in their microbial compositions over the course of the finishing period.”
- Question: Were rumen fluid samples drawn from all animals? Explain this.
- Answer: Yes, rumen fluid samples were drawn from all animals and then the 5 most extremes in efficiency rankings, based on RFI, were selected for analysis after the end of the feedlot period. Lines 82-85 explains the selection process.
- Comment: Separate the Sequencing data part from the statistical analysis part.
- Answer: Thank you, the change has been made.
- Comment: The hypothesis mentioned in line 65 regarding the study is intuitive, so what is new that the study offers?
- Answer: While the basis of the project regarding changes in the microbiome is intuitive, we feel as if this study offers findings that are very relevant to beef cattle production at the feedlot phase. Since we took rumen samples at the beginning and end of the feedlot period, unlike most studies only taking one sample at one point in time, we can compare the developmental changes of the ruminal microbial population when cattle are switched to a grain diet. Additionally, the correlation of feed efficiency and ruminant methanogenesis is very relevant to improving the cattle industry economically and environmentally.
- Question: What were the weights of the animals that started the experiment, what were their weights at the end of the experiment, and were they affected by the final result?
- Answer: The average weight at the beginning of the feedlot period was approximately 512 kg; at the end of that period, it was ~ 597 kg. Most likely not, given that all animals received the exact same diet.
- We thank the reviewer for their kind words and the effort they put in to improving this manuscript.
Reviewer 3 Report
Comments and Suggestions for Authors
Manuscript title: Ruminal Microbiome Differences in Angus Steers with Differing Feed Efficiencies During the Feedlot-Finishing Phase
In a broader context, this study adds to a large body of work on feeding of regimen for bulls to gain maximal growth advantages During the Feedlot-Finishing Phase. Some aspects of the study are not necessarily novel, others, however, are more innovative. These aspects they do improve our understanding of the on microbial compositions in cattle fed during finishing. In general, the subject matter of the article is fascinating and aligns well with the scope of the journal. However, in its current form, it has several shortcomings.
More generally
My main concerns are with the statistical elaboration, which needs to be better described, and of consequence the tables. In fact, dv/sem are missing in the tables, and they need to be set up better to make the reader understand the statistics and the effects of group (high-RFI or low-RFI )and time (beginning (d0) and end (d82) of the feedlot period). A limitation of the study may be the sample size.
Did the authors carry out a specific test to verify whether the sample size is sufficient to explain these types of effects without inducing type I or II errors? If the experiment is underpowered, the authors must clearly state that this is a significant limitation of this experiment.
The abstract effectively summarizes the research content, but the paper would benefit from addressing the identified weaknesses to enhance its impact and applicability.
The literature review provides a good background, but the methods section lacks detailed information on certain aspects of the experimental design and procedures. The conclusions are consistent with the evidence presented, and the references appear to be appropriate but some references need to be checked.
Specific comments
L43-45 Please rewrite this concept by explaining it better
¯ L46 Verify that reference number 8 is suitable and correct for this statement.
¯ L 51-52 Attention: a diet rich in grains and starch needs to be carefully managed because it could have negative implications for the animal's health (e.g., rumnal pH, etc.). The authors should therefore include a few lines on this aspect as well.
¯ L 58 In general, the authors should include some information in the introduction about what other factors may influence feed efficiency and average daily residual gain in cattle.
¯ L82 Please enter the age of the animals
¯ L88 I assume that GrowSafe is a validated system for these operations. Please include some references of validation studies or studies that use it.
¯ L89 Please enter the instrument used for animal body weight detection.
¯ How was the animal housing?
¯ L94 add a space between text and table
¯ Table 1: Please insert dietary energy; delete the row with the total and delete the final blank line.
¯ L97 The ruminal fluid sampling method you mentioned concerns the sampling of very young calves still in the weaning phase. So was the process for harvesting your steers the same?
¯ L99-100 ..."DNA was extracted from 99 rumen fluid samples according to a semi-automated extraction protocol [21]...." reference 21 concerns poultry. I therefore doubt that they studied rumen fluid in this study. Please correct the source or rewrite the sentence to explain it better.
¯ L116 For VFA, was the same ruminal fluid sample used for DNA extraction and sequencing? or were 2 different aliquots taken? Please clarify in the manuscript this aspect.
¯ Have you measured rumen pH?
¯ L140 Authors should more fully describe the statistical analysis; For example: How were comparisons made between pairs of groups? Which multiple comparisons test did you use? Have you considered the interaction?
¯ L153 add a space between text and table
¯ Check the layout of the caption in Table 2.
¯ In table 2 insert body weight data and above all the standard deviation/sem is missing
¯ Table 3 is missing the standard deviation/sem
¯ L160 and table 3 Regarding the results of volatile fatty acids, the authors only report the statistics relating to time (beginning and end of the experiment) but the data explaining whether there were significant differences in the VFA profile between the 2 groups (high-RFI or low-RFI) are missing. In my opinion this is an important aspect and should be included.
¯ Table 4 is missing the standard deviation/sem.
¯ Table 5 and 6. Same thing. Missing dv/sem, missing interaction between groups (high-RFI or low-RFI) and time (days).
¯ In Figure 1 can you improve the graphics of the lettering? (better definition, larger fonts)
¯ L227: Please describe this result better in the description of the results
¯ L358 ensure that the reference list complies with the journal's guidelines.
Author Response
R
In a broader context, this study adds to a large body of work on feeding of regimen for bulls to gain maximal growth advantages During the Feedlot-Finishing Phase. Some aspects of the study are not necessarily novel, others, however, are more innovative. These aspects they do improve our understanding of the on microbial compositions in cattle fed during finishing. In general, the subject matter of the article is fascinating and aligns well with the scope of the journal. However, in its current form, it has several shortcomings.
More generally
My main concerns are with the statistical elaboration, which needs to be better described, and of consequence the tables. In fact, dv/sem are missing in the tables, and they need to be set up better to make the reader understand the statistics and the effects of group (high-RFI or low-RFI) and time (beginning (d0) and end (d82) of the feedlot period). A limitation of the study may be the sample size.
- Question: Did the authors carry out a specific test to verify whether the sample size is sufficient to explain these types of effects without inducing type I or II errors? If the experiment is underpowered, the authors must clearly state that this is a significant limitation of this experiment.
- Answer: Thank you for your comments. Unfortunately, power tests were not conducted; however, you are very correct that this is a significant limitation of this experiment; thus, it has now been stated in the conclusion.
The abstract effectively summarizes the research content, but the paper would benefit from addressing the identified weaknesses to enhance its impact and applicability.
The literature review provides a good background, but the methods section lacks detailed information on certain aspects of the experimental design and procedures. The conclusions are consistent with the evidence presented, and the references appear to be appropriate but some references need to be checked.
Specific comments
- Comment: L43-45 Please rewrite this concept by explaining it better
- Answer: Thank you, the change has been made to expand on the concept.
- Comment: L46 Verify that reference number 8 is suitable and correct for this statement.
- Answer: Thank you for you concern, we checked the claims stated in line 46 with the reference and they were cited correctly. This is the doi:10.2527/jas.2009-2041 of the cited paper, so you can double check if it is suitable.
- Comment: L 51-52 Attention: a diet rich in grains and starch needs to be carefully managed because it could have negative implications for the animal's health (e.g., ruminal pH, etc.). The authors should therefore include a few lines on this aspect as well.
- Answer: Thank you, that is a very good point to add. Information on ruminal acidosis has been added.
- Comment: L 58 In general, the authors should include some information in the introduction about what other factors may influence feed efficiency and average daily residual gain in cattle.
- Answer: Thank you, additional influential factors have been added.
- Comment: L82 Please enter the age of the animals
- Answer: Thank you, the age was added.
- Comment: L88 I assume that GrowSafe is a validated system for these operations. Please include some references of validation studies or studies that use it.
- Answer: Yes, GrowSafe is a well validated system nationwide for feed intake monitoring. We have added three different citations to strengthen the reason GrowSafe systems are effective and reliable. For your convenience, the doi links of the sources are:
- https://doi.org/10.2527/jas.2010-3489
- https://doi.org/10.3168/jds.S0022-0302(03)73962-9
- https://doi.org/10.2527/jas.2005-715
- Comment: L89 Please enter the instrument used for animal body weight detection.
- Answer: It was a digital scale.
- Question: How was the animal housing?
- Answer: It was a concrete floor pen that contained free access to water and feed throughout the day.
- Comment: L94 add a space between text and table
- Answer: Thank you, a space was added.
- Comment: Table 1: Please insert dietary energy; delete the row with the total and delete the final blank line
- Answer: Table 1 shows NEm and NEg, which stands for net energy for maintenance and net energy for gain, respectively. Both are indicators of dietary energy.
- Question: L97 The ruminal fluid sampling method you mentioned concerns the sampling of very young calves still in the weaning phase. So was the process for harvesting your steers the same?
- Answer: Yes, through our own previous research experiences, we have found that no matter the age or the size of the cattle, the sampling method and equipment is the same.
- Comment: L99-100 ..."DNA was extracted from 99 rumen fluid samples according to a semi-automated extraction protocol[21]...." reference 21 concerns poultry. I therefore doubt that they studied rumen fluid in this study. Please correct the source or rewrite the sentence to explain it better.
- Answer: Thank you, yes they did not study rumen fluid in this study. However, the DNA extraction protocol that we utilize is adapted from this study. No matter what the sample is, unless the sample has low bacterial density, we utilize the adaption of the QIAamp DNA Stool Mini Kit, which uses mechanical and enzymatic approaches, as described in the study cited. We have added this clarification to the sentence to explain it better.
- Question: L116 For VFA, was the same ruminal fluid sample used for DNA extraction and sequencing? or were 2 different aliquots taken? Please clarify in the manuscript this aspect.
- Answer: The same ruminal fluid sample was used for both the DNA extraction and VFA analysis. Thank you for pointing out the need for this clarification, the manuscript has been updated to include a clarification sentence of the samples.
- Question: Have you measured rumen pH?
- Answer: No, not for this trial unfortunately.
- Question: L140 Authors should more fully describe the statistical analysis; For example: How were comparisons made between pairs of groups? Which multiple comparisons test did you use? Have you considered the interaction?
- Answer: We added more information regarding statistics to the text. We used Fisher's LSD method after the ANOVA. Fisher's LSD method uses the individual error rate and number of comparisons to calculate the simultaneous confidence level for all confidence intervals. This simultaneous confidence level is the probability that all confidence intervals contain the true difference.
- Comment: L153 add a space between text and table
- Answer: Thank you, a space has been added.
- Comment: Check the layout of the caption in Table 2.
- Answer: Thank you, the layout has been fixed.
- Comment: In table 2 insert body weight data and above all the standard deviation/sem is missing
- Answer: Thank you. This has been fixed.
- Comment: Table 3 is missing the standard deviation/sem
- Answer: Thank you. This has been fixed.
- Comment: L160 and table 3 Regarding the results of volatile fatty acids, the authors only report the statistics relating to time (beginning and end of the experiment) but the data explaining whether there were significant differences in the VFA profile between the 2 groups (high-RFI or low-RFI) are missing. In my opinion this is an important aspect and should be included.
- Comment: Table 4 is missing the standard deviation/sem.
- Answer: Thank you. This has been fixed.
- Comment: Table 5 and 6. Same thing. Missing dv/sem, missing interaction between groups (high-RFI or low-RFI) and time (days).
- Answer: We did not calculate the interactions in this study.
- Question: In Figure 1 can you improve the graphics of the lettering? (better definition, larger fonts)
- Answer: Yes, we increased the font size in this new version of the manuscript.
- Comment: L227: Please describe this result better in the description of the results
- Answer: Thank you, the result is now described more thoroughly.
- Comment: L358 ensure that the reference list complies with the journal's guidelines.
- Answer: Thank you, the guidelines for references for the Microorganisms journal states that the “references may be in any style, provided that you use the consistent formatting throughout.”
- Answer: Yes, GrowSafe is a well validated system nationwide for feed intake monitoring. We have added three different citations to strengthen the reason GrowSafe systems are effective and reliable. For your convenience, the doi links of the sources are:
Round 2
Reviewer 3 Report
Comments and Suggestions for Authors
The authors have significantly improved the manuscript compared to the first version. However, the statistics and results section could be further improved. In fact, as can be seen from the new document, no changes have been made in these sections. Please review my comments and add the indicated corrections into the manuscript.
Author Response
Reviewer 3:
- We thank the reviewer for their kind words and the effort they put in to improving this manuscript. We are sorry that their comments were mixed up in the first response. Their comments helped us immensely in fixing the MS, but I failed in my uploading responses properly.
The authors have significantly improved the manuscript compared to the first version. However, the statistics and results section could be further improved. In fact, as can be seen from the new document, no changes have been made in these sections. Please review my comments and add the indicated corrections into the manuscript.
In a broader context, this study adds to a large body of work on feeding of regimen for bulls to gain maximal growth advantages During the Feedlot-Finishing Phase. Some aspects of the study are not necessarily novel, others, however, are more innovative. These aspects they do improve our understanding of the on microbial compositions in cattle fed during finishing. In general, the subject matter of the article is fascinating and aligns well with the scope of the journal. However, in its current form, it has several shortcomings.
More generally
My main concerns are with the statistical elaboration, which needs to be better described, and of consequence the tables. In fact, dv/sem are missing in the tables, and they need to be set up better to make the reader understand the statistics and the effects of group (high-RFI or low-RFI) and time (beginning (d0) and end (d82) of the feedlot period). A limitation of the study may be the sample size.
- Question: Did the authors carry out a specific test to verify whether the sample size is sufficient to explain these types of effects without inducing type I or II errors? If the experiment is underpowered, the authors must clearly state that this is a significant limitation of this experiment.
- Answer: Thank you for your comments. Unfortunately, power tests were not conducted; however, you are very correct that this is a significant limitation of this experiment; thus, it has now been stated in the conclusion.
The abstract effectively summarizes the research content, but the paper would benefit from addressing the identified weaknesses to enhance its impact and applicability.
The literature review provides a good background, but the methods section lacks detailed information on certain aspects of the experimental design and procedures. The conclusions are consistent with the evidence presented, and the references appear to be appropriate but some references need to be checked.
Specific comments
- Comment: L43-45 Please rewrite this concept by explaining it better
- Answer: Thank you, the change has been made to expand on the concept.
- Comment: L46 Verify that reference number 8 is suitable and correct for this statement.
- Answer: Thank you for you concern, we checked the claims stated in line 46 with the reference and they were cited correctly. This is the doi:10.2527/jas.2009-2041 of the cited paper, so you can double check if it is suitable.
- Comment: L 51-52 Attention: a diet rich in grains and starch needs to be carefully managed because it could have negative implications for the animal's health (e.g., ruminal pH, etc.). The authors should therefore include a few lines on this aspect as well.
- Answer: Thank you, that is a very good point to add. Information on ruminal acidosis has been added.
- Comment: L 58 In general, the authors should include some information in the introduction about what other factors may influence feed efficiency and average daily residual gain in cattle.
- Answer: Thank you, additional influential factors have been added.
- Comment: L82 Please enter the age of the animals
- Answer: Thank you, the age was added.
- Comment: L88 I assume that GrowSafe is a validated system for these operations. Please include some references of validation studies or studies that use it.
- Answer: Yes, GrowSafe is a well validated system for feed intake monitoring. We have added three different citations to strengthen the reason GrowSafe systems are effective and reliable. For your convenience, the doi links of the sources are:
- https://doi.org/10.2527/jas.2010-3489
- https://doi.org/10.3168/jds.S0022-0302(03)73962-9
- https://doi.org/10.2527/jas.2005-715
- Comment: L89 Please enter the instrument used for animal body weight detection.
- Answer: It was a digital scale.
- Question: How was the animal housing?
- Answer: It was a concrete floor pen that contained free access to water and feed throughout the day.
- Comment: L94 add a space between text and table
- Answer: Thank you, a space was added.
- Comment: Table 1: Please insert dietary energy; delete the row with the total and delete the final blank line
- Answer: Table 1 shows NEm and NEg, which stands for net energy for maintenance and net energy for gain, respectively. Both are indicators of dietary energy.
- Question: L97 The ruminal fluid sampling method you mentioned concerns the sampling of very young calves still in the weaning phase. So was the process for harvesting your steers the same?
- Answer: Yes, through our own previous research experiences, we have found that no matter the age or the size of the cattle, the sampling method and equipment is the same.
- Comment: L99-100 ..."DNA was extracted from 99 rumen fluid samples according to a semi-automated extraction protocol[21]...." reference 21 concerns poultry. I therefore doubt that they studied rumen fluid in this study. Please correct the source or rewrite the sentence to explain it better.
- Answer: Thank you, yes they did not study rumen fluid in this study. However, the DNA extraction protocol that we utilize is adapted from this study. No matter what the sample is, unless the sample has low bacterial density, we utilize the adaption of the QIAamp DNA Stool Mini Kit, which uses mechanical and enzymatic approaches, as described in the study cited. We have added this clarification to the sentence to explain it better.
- Question: L116 For VFA, was the same ruminal fluid sample used for DNA extraction and sequencing? or were 2 different aliquots taken? Please clarify in the manuscript this aspect.
- Answer: The same ruminal fluid sample was used for both the DNA extraction and VFA analysis. Thank you for pointing out the need for this clarification, the manuscript has been updated to include a clarification sentence of the samples.
- Question: Have you measured rumen pH?
- Answer: No, not for this trial unfortunately.
- Question: L140 Authors should more fully describe the statistical analysis; For example: How were comparisons made between pairs of groups? Which multiple comparisons test did you use? Have you considered the interaction?
- Answer: We added more information regarding statistics to the text. We used Fisher's LSD method after the ANOVA. Fisher's LSD method uses the individual error rate and number of comparisons to calculate the simultaneous confidence level for all confidence intervals. This simultaneous confidence level is the probability that all confidence intervals contain the true difference.
- Comment: L153 add a space between text and table
- Answer: Thank you, a space has been added.
- Comment: Check the layout of the caption in Table 2.
- Answer: Thank you, the layout has been fixed.
- Comment: In table 2 insert body weight data and above all the standard deviation/sem is missing
- Answer: Thank you. This has been fixed.
- Comment: Table 3 is missing the standard deviation/sem
- Answer: Thank you. This has been fixed.
- Comment: L160 and table 3 Regarding the results of volatile fatty acids, the authors only report the statistics relating to time (beginning and end of the experiment) but the data explaining whether there were significant differences in the VFA profile between the 2 groups (high-RFI or low-RFI) are missing. In my opinion this is an important aspect and should be included.
- Answer: Thank you. This has been fixed.
- Comment: Table 4 is missing the standard deviation/sem.
- Answer: Thank you. This has been fixed.
- Comment: Table 5 and 6. Same thing. Missing dv/sem, missing interaction between groups (high-RFI or low-RFI) and time (days).
- Answer: We did not calculate the interactions in this study.
- Question: In Figure 1 can you improve the graphics of the lettering? (better definition, larger fonts)
- Answer: Yes, we increased the font size in this new version of the manuscript.
- Comment: L227: Please describe this result better in the description of the results
- Answer: Thank you, the result is now described more thoroughly.
- Comment: L358 ensure that the reference list complies with the journal's guidelines.
- Answer: Thank you, the guidelines for references for the Microorganisms journal states that the “references may be in any style, provided that you use the consistent formatting throughout.”